# Mental Wellbeing and Health-Risk Behaviours of University Students in Brunei: A Cross-Sectional Study during COVID-19 Pandemic

**DOI:** 10.3390/healthcare11162327

**Published:** 2023-08-17

**Authors:** Hanif Abdul Rahman, Nurul Nazurah Julaini, Siti Nurzaimah Nazhirah Zaim, Nurfatin Amalina Masri, Khadizah H. Abdul-Mumin

**Affiliations:** 1PAPRSB Institute of Health Sciences, Universiti Brunei Darussalam, Gadong BE1410, Brunei; 22m1832@ubd.edu.bn (N.N.J.); 21m8731@ubd.edu.bn (S.N.N.Z.); 22m1831@ubd.edu.bn (N.A.M.); khadizah.mumin@ubd.edu.bn (K.H.A.-M.); 2School of Nursing and Statistics Online Computational Resource, University of Michigan, Ann Arbor, MI 48109, USA

**Keywords:** COVID-19, health-risk behaviors, mental wellbeing, university students

## Abstract

**Background**: The coronavirus disease discovered in 2019 (COVID-19) has impacted the health behaviour of university students in a range of western nations and had a number of detrimental effects on their mental wellbeing. However, little is known about the impact on university students in Southeast Asia, particularly in Brunei. This study aims to estimate the prevalence and factors related to poor mental wellbeing and engaging in health-risk behaviors among university students in Brunei during the COVID-19 pandemic. **Methods**: A cross-sectional online survey of students in all public universities was conducted from March to June 2021. Warwick-Edinburgh Mental Well-being Scale, Global Physical Activity Questionnaire, and WHO STEPs questionnaires were used to measure mental wellbeing, physical activity, and health behaviors, respectively. Logistic regressions were applied. **Results**: A total of 1020 university students participated in this study (57% response rate). Prevalence of poor mental wellbeing and physical inactivity were 30% and 42.8%, respectively. Excessive snacking (89.7%), sugar intake (81.7%), and salt intake (53.8%) were the most prevalent health-risk behaviors. Body mass index and participating in sports activities were the most common factors related to mental wellbeing and health-risk behaviors. **Conclusions**: Participating in certain sports activities appears to have protective effects against study outcomes. However, a high prevalence of physical inactivity and unhealthy diet among university students is a concern, as it could be indicative of future non-communicable disease trends.

## 1. Introduction

The primary focus in the fight against the coronavirus disease (COVID-19) pandemic of healthcare policies implemented by local and national governments was on the detrimental effects on physical health [1]. The effects on psychological health are often considered secondary concerns, however, many studies have shown that COVID-related prevention measures of partial or nationwide lockdown have moderate to severe psychological and emotional impacts on university students [2,3]. Experiences of fear, hopelessness, and a profound loss of purpose due to the pandemic have increased the prevalence of generalized anxiety disorder, depression, and stress [4]. Additionally, in this age of social media, widespread rumors and misinformation have also exacerbated mental status of the general population, including university students [1].

Most countries enacted strategies involving compulsory confinement or quarantine and restrictions on free movement that have likely changed lifestyle health behaviors of university students [5]. Studies varied in the effects of the impact of the COVID-19 pandemic on health-risk behaviors of university students, such as alcohol consumption, smoking, physical inactivity, and unhealthy diet [5,6,7]. Although these behaviors among university students are commonly reported in many regions, limited research has reported the prevalence of these health-risk behaviors in Southeast Asian universities, particularly, during the COVID-19 pandemic. Therefore, this study aimed to establish a baseline estimate of the prevalence of poor mental wellbeing and health-risk behaviors among university students. This is a part of a larger ASEAN University Network-Health Promotion Network (AUN-HPN) study from seven Southeast Asian countries collected from March to June 2021, when COVID-19 restrictions in different forms were enforced in all universities. This report is specific to Brunei. The aim of this paper is to report the prevalence and factors related to poor mental wellbeing and health-risk behaviors of university students in Brunei.

## 2. Materials and Methods

### 2.1. Study Design and Procedure

Details of the AUN-HPN study procedures are reported elsewhere [8]. Briefly, this is a cross-sectional online survey that was distributed to all public universities in Brunei. All university students were eligible to participate, and they completed a self-administered questionnaire in Qualtrics.

### 2.2. Research Instruments

The survey parameters were measured using several validated questionnaires. The shortened Warwick-Edinburgh Mental Well-being Scale (WEMWBS) was used to measure mental wellbeing [9]. WEMWBS is a 7-item scale of mental wellbeing, covering subjective wellbeing and psychological functioning, in which all items are worded positively and address aspects of positive mental health. The scale is scored by summing responses to each item answered on a 1 to 5 Likert scale (1 = None of the time, 5 = All of the time). The minimum score is 7 and the maximum is 35. Those scored between 7.0 and 17.99 were considered as having poor mental wellbeing, and 18.0 and above was considered good mental wellbeing.

Physical activity (PA) was measured by the Global Physical Activity Questionnaire (GPAQ) [10]. PA was expressed as a Metabolic Equivalent of Task unit (MET-units/week), and those students who achieved a minimum of 600 MET-week were considered physically active. Other health-risk behaviors, including smoking, alcohol consumption, diet intake, salt intake, and sugar intake, were measured using the 16-item WHO STEPs survey [11]. For smoking, students were asked about their smoking status (smoke daily/smoke occasionally/do not smoke now, but used to/tried smoking a few times, but never smoked regularly/I have never smoked). Students who smoked daily were categorized into “current smokers” and other responses were collapsed into “not current smokers”. For alcohol consumption, students were asked how many days in a week did they usually drink alcohol. Response options ranged from 0–7 days and “don’t ever drink”. Students’ drinking was classified as “daily”, if they drank 7 days/week, and “not daily” for responses of <7 days/week. For fruit/vegetable consumption, students were asked how many servings of fruits/vegetable they usually ate each day (1 to 8 servings). Students were classified into “Sufficient” (≥5 servings/day), and “Insufficient” (<5 servings/day).

### 2.3. Data Analysis

Weighted probability was applied to adjust and compensate for non-response bias. Descriptive statistics were used to describe the sample characteristics. The Chi-square test was applied to establish gender-stratified comparisons of mental wellbeing and health-risk behavior level. Logistic regression was computed to estimate the factors related to poor mental wellbeing and negative health behaviors. All statistical operations were conducted with R v4.1.1.

### 2.4. Ethical Considerations

The study protocol was reviewed and approved by the university research and ethics committee in accordance with relevant local and international ethical guidelines, such as Declarations of Helsinki (UBD/OAVCR/UREC/OCTOBER2020-05). All students who participated in the study provided their online informed consent.

## 3. Results

The study sample comprised of 1020 university students (response rate = 57%), where 65% were between the age of 18 to 21, and the remaining were over 21 years old. The proportion of female students was 64% and male students 36%, and the majority were living off-campus (89%) and in single houses (77%). First-class students with a grade point average above 3.9 was at 11%. About one third (30%) were either overweight or obese. Regarding sleeping duration, 70% slept less than seven hours per day, and 40% reported more than 8 h of daily sedentary activity. Running or jogging (58%) was the most common sport activity, followed by aerobic dance (41%), racquet sports (24%), ball-related sports (23%), and water sports (21%) (see Table 1).

Figure 1 presents the gender-stratified prevalence of poor mental wellbeing and health-risk behaviors among the sample of university students. The overall prevalence of poor mental wellbeing was 30%, where female students (33.6%) reported significantly higher than male students (23.4%) (*p* < 0.001). Eating excessive snacks was the most prevalent health-risk behavior for both female (92.4%) and male (84.8%) students. Female students were reportedly less physically active (46.4%) and had a higher proportion of alcohol drinkers (4.6%) than male students, whereas smoking among male students (27.0%) was significantly higher than female students (5.8%) (*p* < 0.001).

Table 2 and Table 3 demonstrate the final adjusted multiple logistic regression models for factors related to poor mental wellbeing and health-risk behaviors. After adjusting for confounders, poor mental wellbeing was significantly more likely among people who were female (OR = 2.00, 95% CI: 1.41, 2.85), smoking (OR = 2.17, 95% CI: 1.39, 3.38), being overweight (OR = 1.85, 95% CI: 1.12, 2.77) or obese (OR = 1.77, 95% CI: 1.08, 2.98), consuming excessive sweet drinks (OR = 1.58, 95% CI: 1.07, 2.38), and participating in fight sports (OR = 2.74, 95% CI: 1.73, 4.37). Students who did water sports were significantly less likely to experience poor mental wellbeing (OR = 0.28, 95% CI: 0.18, 0.44).

In the final adjusted logistic regression model, there were significantly increased odds of physical inactivity among people who were female (OR = 1.90, 95% CI: 1.37, 2.65), being overweight (OR = 1.81, 95% CI: 1.23, 2.66), and smoking (OR = 1.90, 1.25, 2.91), whereas significantly decreased odds of physical inactivity among people who were alcohol drinkers (OR = 0.26, 95% CI: 0.08, 0.66), consuming excessive sweet drinks (OR = 0.47, 95% CI: 0.30, 0.75), and participating in running/jogging (OR = 0.42, 95% CI: 0.32, 0.56), aerobic dance (OR = 0.46, 95% CI: 0.34, 0.61) or ball-related sports (OR = 0.53, 95% CI: 0.36, 0.77).

In the final adjusted model, there were significantly higher odds of smoking among those who were overweight (OR = 2.18, 95% CI: 1.29, 3.70) or obese (OR = 3.04, 95% CI: 1.56, 5.83), sleeping less than 7 h per day (OR = 1.46, 95% CI: 1.07, 2.00), consuming excessive snacks (OR = 2.81, 95% CI: 1.34, 6.46), experiencing poor mental wellbeing (3.08, 95% CI: 1.93, 4.97), being physically inactive (OR = 1.75, 95% CI: 1.10, 2.78), and participating in water sports (OR = 3.55, 95% CI: 1.99, 6.38) and fight sports (OR = 4.55, 95% CI: 2.56, 8.18), whereas there was significantly lower odds of smoking among females, and those who participated in aerobic dance, ball-related sports, or racquet sports.

In the final adjusted model, being female (OR = 34.6, 95% CI: 7.35, 226.0), smoking (OR = 23.9, 95% CI: 4.77, 124.0), and participating in racquet sports (OR = 9.67, 95% CI: 3.32, 31.7) or fight sports (OR = 4.55, 95% CI: 2.56, 8.18) were associated with increased odds of alcohol consumption, whereas significantly higher odds of poor diet intake were associated with being underweight (OR = 4.25, 95% CI: 2.81, 6.46) or obese (OR = 2.63, 95% CI: 1.64, 4.20), consuming alcohol (OR = 3.00, 95% CI: 1.39, 6.91), and participating in running/jogging, cycling, ball-related sports, or fight sports.

Significantly higher odds of excessive sugar intake were associated with being underweight (OR = 4.44, 95% CI: 2.28, 9.59) or obese (OR = 4.63, 2.28, 10.4), smoking (OR = 1.73, 95% CI: 1.01, 3.07), excessive snack intake (OR = 4.31, 95% CI: 2.66, 7.01) and salt intake (OR = 2.01, 95% CI: 1.40, 2.88), and participating in aerobic dance (OR = 1.51, 95% CI: 1.03, 2.23) or racquet sports (OR = 2.89, 95% CI: 1.78, 4.86). Those who participated in running/jogging had lower odds of excessive sugar intake.

Excessive snack intake was significantly more likely among smokers and those who consumed high salt and high sugar intake. Students who were obese and alcohol drinkers were less likely to consume excessive snacks. Meanwhile, excessive salt intake was significantly more likely among those who sleep less than 7 h per day, consume excessive snacks, and have high sugar intake; however, it was less likely for those doing aerobic dance exercise.

## 4. Discussion

Overall, the prevalence of poor mental wellbeing (30%) was comparatively lower than previous studies during the pandemic from China (45.1%), the United Kingdom (40.5%), France (42.8%), and the United States (45.0%) [12,13,14,15]. We suspect the lower prevalence might be underreported due to stigma related to mental illness in Asian cultures [16]. However, social capital is an important asset in Asian families and communities that could have protective effects on the students’ health and wellbeing [17]. Family support could also be an underlying reason for the lower prevalence of poor mental wellbeing. The international lockdown measures, such as school closures, that were implemented to mitigate the spread of COVID-19 have led to an increase in social isolation [18], which has been linked to increases in depression, anxiety, and suicidal ideation [19,20]. Unlike other countries, the majority of university students in Brunei are living off-campus. This trend is common in the country, as most students are living with their families while in university. Family support is paramount to students during COVID-19, as it covers environmental support, emotional support, and capability support [21]. Nevertheless, the Asian culture of stigmatizing and discriminating against those with mental issues needs to be addressed, as it could become a recovery barrier and hamper recovery post-COVID-19 [22].

In terms of health-risk behaviors, there are particular concerns about the high prevalence of physical inactivity and unhealthy diet, such as high consumption of salt, snacks, and sugar. These unhealthy lifestyle choices are major risk factors for the development of non-communicable diseases (NCDs) in the future [23]. NCDs such as cardiovascular disease and diabetes are the top causes of mortality and morbidity in Brunei [24]. Higher proportions of alcohol consumption among female students during COVID-19 were similar to a cross-sectional study in Germany [6]. This could be due to a higher percentage of poor mental wellbeing and poor diet; particularly, increased consumption of alcohol and sugary foods have been linked to experiences of psychological distress [25,26,27]. Engagement in physical activity can mediate stress response and promote mental wellbeing. It has been demonstrated that physical activity has a positive impact on symptoms of anxiety and depression. Women tend to be less physically active than men; therefore, they experience higher levels of psychological distress [28].

Next, the prevalence of high sedentary duration among students is not surprising as the university shifted from face-to-face to remote teaching and learning during COVID-19. This situation favors many factors that encourage sedentary behaviors, including intentions to be sedentary, residency, prolonged screen time from computer and TV access, and university workloads, such as assignments and exams [29]. Specifically, students spent less time in physical education class and transportation [30].

The current lifestyle behavior of students at university provides a good forecast in the NCD levels when they reach adulthood. Therefore, it is crucial that university and related stakeholders intervene and reduce these unhealthy behaviors, as they could continue them in the transitionary phase from adolescence to adulthood later in life.

Specifically, this study has found salient factors related to poor mental wellbeing and health-risk behaviors that should be taken into account when formulating strategies and interventions to promote a healthy lifestyle and minimize the risk of developing NCDs in the future. Particularly, the monitoring of body mass index and encouraging students to take up sport activities such as water sports, aerobic exercise, or running/jogging offers beneficial effects and could lower odds of negative health behaviors and poor mental wellbeing.

Further to improving mental wellbeing, practical ways have been studied on short- and long-term effects of wellbeing interventions after the COVID-19 pandemic that students could explore, such as the seven-week self-managed activities [31] and increasing gratitude [32].

### Study Limitations

This study was limited by its cross-sectional design where statistical inference on causal and interactions are limited. Measures using self-reported online questionnaires could underestimate certain health-risk behaviors such as smoking and alcohol consumption. Nonetheless, this study provides a unique epidemiological estimate of the poor mental wellbeing and health-risk behaviors seen during the pandemic. Future prospective and interventional study designs could examine the effects against the baseline results from this study and provide more robust evidence for university policies in the post-pandemic era.

## 5. Conclusions

The present study extends previous results from other countries reporting the prevalence and factors associated with poor mental wellbeing and health-risk behaviors among university students during the COVID-19 pandemic. When students re-enter the academic institutions, it is important to re-examine and continue to monitor students’ health behaviors, particularly physical activity and body mass index, to provide a targeted health promotion intervention in the university to reduce the risk of developing non-communicable diseases in the future.

## Figures and Tables

**Figure 1 healthcare-11-02327-f001:**
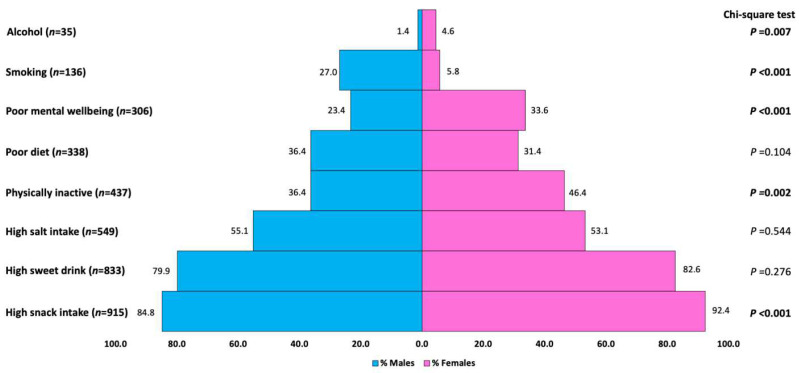
Prevalence of Health-risk behaviors and Mental Wellbeing of University students in Brunei (*n* = 1020).

**Table 1 healthcare-11-02327-t001:** Sample characteristics of university students (*n* = 1020).

	Frequency (Percentage)
**Age groups**	
18 years old	111 (11%)
19 to 21 years old	549 (54%)
More than 21 years old	360 (35%)
**Gender**	
Male	363 (36%)
Female	657 (64%)
**Body Mass Index (BMI)**	
Normal weight	561 (55%)
Underweight	155 (15%)
Overweight	194 (19%)
Obese	110 (11%)
**Grade point average (GPA) (Range 1 to 5)**	
>3.9 (First class)	110 (11%)
3.2 to 3.9 (Upper second class)	465 (45%)
≤3.2 (Lower second and third class)	445 (44%)
**Living arrangement**	
On-campus	113 (11%)
Off-campus	907 (89%)
**Housing type**	
Single house	784 (77%)
Townhouse	170 (17%)
Apartment	66 (6.5%)
**Sedentary duration (hours)**	
≤3 h	236 (23%)
4 to 8 h	381 (37%)
>8 h	403 (40%)
**Sleeping duration (hours)**	
≤7 h	710 (70%)
>7 h	310 (30%)
**Type of sport activities**	
Running/Jogging	587 (58%)
Cycling	177 (17%)
Aerobic dance	416 (41%)
Ball-related sport	231 (23%)
Racquet sport	243 (24%)
Athletics	63 (6.2%)
Water sport	211 (21%)
Fight sport	145 (14%)

**Table 2 healthcare-11-02327-t002:** Multiple logistic regression results on factors related to smoking, alcohol, physical inactivity, and mental wellbeing (*n* = 1020).

	Smoking			Alcohol			Physically Inactive			Poor Mental Wellbeing		
Factors	OR	95% CI	*p*-Value	OR	95% CI	*p*-Value	OR	95% CI	*p*-Value	OR	95% CI	*p*-Value
**Gender (Ref: Male)**	1.00	-	-	1.00	-	-	1.00	-	-	1.00	-	-
Female	0.13	0.08, 0.21	**<0.001**	34.6	7.35, 226	**<0.001**	1.90	1.37, 2.65	**<0.001**	2.00	1.41, 2.85	**<0.001**
**BMI (Ref: Normal)**	1.00	-	-	-	-	-	1.00	-	-	1.00	-	-
Underweight	0.44	0.14, 1.11	0.11	-	-	-	1.38	0.92, 2.06	0.12	1.08	0.71, 1.65	0.7
Overweight	2.18	1.29, 3.70	**0.004**	-	-	-	1.81	1.23, 2.66	**0.003**	1.85	1.23, 2.77	**0.003**
Obese	3.04	1.56, 5.83	**<0.001**	-	-	-	0.69	0.43, 1.09	0.11	1.77	1.08, 2.89	**0.024**
**Sleep duration (Ref: >7 h)**	1.00	-	-	-	-	-	-	-	-	-	-	-
≤7 h	1.46	1.07, 2.00	**0.018**	-	-	-	-	-	-	-	-	-
**Health-risk behaviors (Ref: No)**	1.00	-	-	1.00	-	-	1.00	-	-	1.00	-	-
Smoking	-	-	-	23.9	4.77, 124	**<0.001**	1.90	1.25, 2.91	**0.003**	2.17	1.39, 3.38	**<0.001**
Alcohol intake	-	-	-	-	-	-	0.26	0.08, 0.66	**0.009**	-	-	-
High sugar intake	-	-	-	-	-	-	-	-	-	1.58	1.07, 2.38	**0.025**
High snack intake	2.81	1.34, 6.46	**0.010**	-	-	-	0.47	0.30, 0.75	**0.002**	-	-	-
Poor mental wellbeing	3.08	1.93, 4.97	**<0.001**	-	-	-	-	-	-	-	-	-
Physically inactive	1.75	1.10, 2.78	**0.018**	-	-	-	-	-	-	-	-	-
**Types of sport (Ref: No)**	1.00	-	-	-	-	-	1.00	-	-	1.00	-	-
Running/Jogging	-	-	-	-	-	-	0.42	0.32, 0.56	**<0.001**	-	-	-
Cycling	-	-	-	-	-	-	-	-	-	-	-	-
Aerobic dance	0.57	0.35, 0.93	**0.026**	-	-	-	0.46	0.34, 0.61	**<0.001**	-	-	-
Ball-related sport	0.38	0.19, 0.71	**0.003**	-	-	-	0.53	0.36, 0.77	**0.001**	-	-	-
Racquet sport	0.22	0.10, 0.46	**<0.001**	9.67	3.32, 31.7	**<0.001**	-	-	-	-	-	-
Water sport	3.55	1.99, 6.38	**<0.001**	-	-	-	-	-	-	0.28	0.18, 0.44	**<0.001**
Fight sport	4.55	2.56, 8.18	**<0.001**	4.16	1.34, 13.2	**0.014**	-	-	-	2.74	1.73, 4.37	**<0.001**
H-L Goodness of fit test	**11.9 (3)**			**3.7 (8)**			**9.4 (8)**			**18.9 (8)**		
*p*-value	**0.076**			**0.879**			**0.599**			**0.152**		
McFadden R-square	**0.275**			**0.564**			**0.126**			**0.119**		

OR = Odds ratio; CI = Confidence interval; Ref = Reference group; H-L = Hosmer and Lemeshow; Bold values = statistical significance at 0.05 level.

**Table 3 healthcare-11-02327-t003:** Multiple logistic regression results on factors related to high salt intake, high sugar intake, high snack intake, and poor diet (*n* = 1020).

	High Salt Intake			High Sugar Intake			High Snack Intake			Poor Diet		
Factors	OR	95% CI	*p*-Value	OR	95% CI	*p*-Value	OR	95% CI	*p*-Value	OR	95% CI	*p*-Value
**Gender (Ref: Male)**	1.00	-	-	1.00	-	-	1.00	-	-	1.00	-	-
Female	1.07	0.81, 1.43	0.600	0.86	0.58, 1.27	0.400	3.21	2.01, 5.20	**<0.001**	1.16	0.83, 1.64	0.400
**BMI (Ref: Normal)**	-	-	-	1.00	-	-	1.00	-	-	1.00	-	-
Underweight	-	-	-	4.44	2.28, 9.59	**<0.001**	1.06	0.51, 2.40	0.9	4.25	2.81, 6.46	**<0.001**
Overweight	-	-	-	1.01	0.64, 1.63	>0.9	1.09	0.61, 2.03	0.8	1.40	0.94, 2.08	0.100
Obese	-	-	-	4.63	2.28, 10.4	**<0.001**	0.24	0.13, 0.46	**<0.001**	2.63	1.64, 4.20	**<0.001**
**Sleep duration (Ref: >7 h)**	1.00	-	-	-	-	-	-	-	-	-	-	-
≤7 h	1.38	1.04, 1.84	**0.025**	-	-	-	-	-	-	-	-	-
**Health-risk behaviors (Ref: No)**	-	-	-	1.00	-	-	1.00	-	-	1.00	-	-
Smoking	-	-	-	1.73	1.01, 3.07	0.054	2.89	1.41, 6.50	**0.006**	-	-	-
Alcohol intake	-	-	-	-	-	-	0.16	0.06, 0.53	**<0.001**	3.00	1.39, 6.91	**0.007**
High snack intake	1.91	1.24, 2.99	**0.004**	4.31	2.66, 7.01	**<0.001**				-	-	-
High salt intake	-	-	-	2.01	1.40, 2.88	**<0.001**	1.77	1.13, 2.80	**0.013**	-	-	-
High sugar drink	1.88	1.33, 2.66	**<0.001**	-	-	-	5.22	3.21, 8.52	**<0.001**	-	-	-
Poor mental wellbeing	-	-	-	1.76	1.17, 2.69	**0.008**	-	-	-	-	-	-
**Types of sport (Ref: No)**	1.00	-	-	1.00	-	-	-	-	-	1.00	-	-
Running/Jogging	-	-	-	0.36	0.24, 0.53	**<0.001**	-	-	-	1.83	1.33, 2.51	**<0.001**
Cycling	-	-	-	-	-	-	-	-	-	1.70	1.16, 2.48	**0.006**
Aerobic dance	0.77	0.59, 1.01	0.056	1.51	1.03, 2.23	**0.036**	-	-	-	-	-	-
Ball-related sport	-	-	-	-	-	-	-	-	-	3.56	2.51, 5.08	**<0.001**
Racquet sport	-	-	-	2.89	1.78, 4.86	**<0.001**	-	-	-	-	-	-
Fight sport	-	-	-	-	-	-	-	-	-	1.72	1.11, 2.65	**0.015**
H-L Goodness of fit test	**12.6 (8)**			**15.9 (8)**			**7.4 (8)**			**7.2 (8)**		
*p*-value	**0.126**			**0.331**			**0.405**			**0.282**		
McFadden R-square	**0.064**			**0.166**			**0.151**			**0.149**		

OR = Odds ratio; CI = Confidence interval; Ref = Reference group; H-L = Hosmer and Lemeshow, Bold values = statistical significance at 0.05 level.

## Data Availability

Data is available upon reasonable request due to university and ethics restrictions based on data protection protocols.

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
