# Peer review of "Mental Wellbeing and Health-Risk Behaviours of University Students in Brunei: A Cross-Sectional Study during COVID-19 Pandemic"

_healthcare, 2023, doi:10.3390/healthcare11162327_

Round 1
Reviewer 1 Report
The topic of the MS is about the rate of risk behavious in Brunei students. The results are interesting and have important practical implications. However, I would suggest before publicaton
1. Improving the Introduction. Something more should be written about previous researches in the field and a standing alone section 'Aims and hypotheses' provided
2. I understand that details of the instruments are reported elsewhere. However something more should be added here to better grasp what asssessed and how
3. There are some interesting gender differences I would discuss deeply also on the light of previous researches on beliefs and stereotypes (e.g., Moè et al., 2021)
4. I would also add a Practical Implications section stating what can be done to improve student well-being, for instance increasing gratitude (e.g., Buragohain & Mandal, 2015) or overall well-being (e.g., Moè, 2022).
Suggested references
Buragohain, P., & Mandal, R. (2015). Teaching of gratitude among the students of secondary school as a means of well-being. International Journal of Humanities Social Science and Education, 2(2), 179-188.
Moè, A. (2022). Does the weekly practice of recalling and elaborating episodes raise well-being in university students? Journal of Happiness Studies, 23(7), 3389–3406.
Moè, A., Hausmann, M., & Hirnsten, M. (2021). Gender stereotypes and incremental beliefs in STEM and non-STEM students in three countries. Relationships with performance in cognitive tasks. Psychological Research, 85, 554-567.
Author Response
Thank you and we greatly appreciate your valuable time to review and comment on our humble manuscript. Relevant changes according to reviewers’ comments is highlighted in yellow in the revised manuscript, attached.
The topic of the MS is about the rate of risk behavious in Brunei students. The results are interesting and have important practical implications. However, I would suggest before publication.
Response: Thank you for your valuable time to review and comment on our humble manuscript. All changes according to reviewers’ comments is highlighted in yellow in the revised manuscript.
- Improving the Introduction. Something more should be written about previous researches in the field and a standing alone section 'Aims and hypotheses' provided. Response: Reference to past account and literatures have been mentioned. Aim of the study added.
- I understand that details of the instruments are reported elsewhere. However something more should be added here to better grasp what asssessed and how. Response: More details of the instruments, added.
- There are some interesting gender differences I would discuss deeply also on the light of previous researches on beliefs and stereotypes (e.g., Moè et al., 2021). Response: Reference to the article, added.
-
I would also add a Practical Implications section stating what can be done to improve student well-being, for instance increasing gratitude (e.g., Buragohain & Mandal, 2015) or overall well-being (e.g., Moè, 2022).
Suggested references
Buragohain, P., & Mandal, R. (2015). Teaching of gratitude among the students of secondary school as a means of well-being. International Journal of Humanities Social Science and Education, 2(2), 179-188.
Moè, A. (2022). Does the weekly practice of recalling and elaborating episodes raise well-being in university students? Journal of Happiness Studies, 23(7), 3389–3406.
Moè, A., Hausmann, M., & Hirnsten, M. (2021). Gender stereotypes and incremental beliefs in STEM and non-STEM students in three countries. Relationships with performance in cognitive tasks. Psychological Research, 85, 554-567.
Response: Practical statement, added. Suggested references, added.

Reviewer 2 Report
The manuscript addresses a very interesting topic of research. However, some points could be improved:
(i) It is suggested to clarify in the abstract the year of data collection and the group of participants involved;
(ii) In point "2.2 Research instrument" a more detailed clarification of the instruments used is also recommended, as well as a clarification of the variables presented in Table 1.
Author Response
Thank you and we greatly appreciate your valuable time to review and comment on our humble manuscript. Relevant changes according to reviewers’ comments is highlighted in yellow in the revised manuscript, attached.
The manuscript addresses a very interesting topic of research. However, some points could be improved:
(i) It is suggested to clarify in the abstract the year of data collection and the group of participants involved;
(ii) In point "2.2 Research instrument" a more detailed clarification of the instruments used is also recommended, as well as a clarification of the variables presented in Table 1.
Response:
Thank you for your valuable time to review and comment on our humble manuscript. All changes according to reviewers’ comments is highlighted in yellow in the revised manuscript.
(i) Data collection year, added in abstract.
(ii) More details of research instruments, added.

Reviewer 3 Report
I provide here some comments that might improve the quality of the “brief report”.
Regarding the keywords, it is recommended to place them in alphabetical order.
About the Introduction:
It is recommended to update the introduction with data from some more recent publications, since the most current references are only two works from 2022 and none from 2023.
About the Results:
In table 3 of results, review if perhaps data related to the “gender” factor should be included.
Likewise, it would be of interest for the study to incorporate into the results data of average values ​​(means and standard deviation) relative to the sample used, at least from the Mental Well-Being Scale (WEMWBS) measure, as well as normative values ​​and cut-off points of it.
Also, comment in the discussion if the well-being levels of the sample were significantly different or similar with respect to the normative values ​​and cut-off points of the validation data closest to the study sample with that instrument.
Author Response
Thank you and we greatly appreciate your valuable time to review and comment on our humble manuscript. Relevant changes according to reviewers’ comments is highlighted in yellow in the revised manuscript, attached.
I provide here some comments that might improve the quality of the “brief report”.
Response:
Thank you for your valuable time to review and comment on our humble manuscript. All changes according to reviewers’ comments is highlighted in yellow in the revised manuscript.
Regarding the keywords, it is recommended to place them in alphabetical order.
Response: Keywords changed to alphabetical order.
About the Introduction:
It is recommended to update the introduction with data from some more recent publications, since the most current references are only two works from 2022 and none from 2023.
Response: Reference to past account and literatures have been mentioned.
About the Results:
In table 3 of results, review if perhaps data related to the “gender” factor should be included.
Response: Results of Gender in Table 3, added
Likewise, it would be of interest for the study to incorporate into the results data of average values ​​(means and standard deviation) relative to the sample used, at least from the Mental Well-Being Scale (WEMWBS) measure, as well as normative values ​​and cut-off points of it.
Response: Mental wellbeing is a categorical variable - poor and good. Result is expressed as frequency and percentage.
Also, comment in the discussion if the well-being levels of the sample were significantly different or similar with respect to the normative values ​​and cut-off points of the validation data closest to the study sample with that instrument.
Response: Discussion on mental wellbeing have been mentioned and expanded.

Reviewer 4 Report
The paper provides an interesting contribution to the literature pertaining to the impact of COVID 19, however the initial foregrounding is a little too cursory and the reviewer recommends that further elaboration be included to ensure the readers are able to see the value in the study, the aims of the study, the existing literature that positions the study and the concepts which are critical to the study. Further, there needs to be a baseline presented, given the population are a group which is likely to engage in health risks such as smoking and drinking, a base line is required in order to illustrate change related to COVID. Alson consider explaining any cultural context relevant to your study and population.
The initial introduction could benefit from a little deeper review of the literature that already exist and how this study contributes or addresses a gap. Also, define the terminology crucial to the study, what are health risk behaviours for example, as various disciplines will interpret this differently.
The methods section needs to provide more specific information about the measurement, and exactly what is explored, the paper gives examples of the behaviour, but the paper should clearly articulate all the behaviours explored. Research instruments should be explained in detail.
Also, why did you focus on university students? there should be some justification of methodological choices made.
This paper would benefit greatly from some comparison between pre and post covid statistics.
Author Response
Thank you and we greatly appreciate your valuable time to review and comment on our humble manuscript. Relevant changes according to reviewers’ comments is highlighted in yellow in the revised manuscript, attached.
The initial introduction could benefit from a little deeper review of the literature that already exist and how this study contributes or addresses a gap. Also, define the terminology crucial to the study, what are health risk behaviours for example, as various disciplines will interpret this differently.
Response: Thank you for your valuable time to review and comment on our humble manuscript. All changes according to reviewers’ comments is highlighted in yellow in the revised manuscript. Reference to past account and literatures have been mentioned.
The methods section needs to provide more specific information about the measurement, and exactly what is explored, the paper gives examples of the behaviour, but the paper should clearly articulate all the behaviours explored. Research instruments should be explained in detail.
Response: More details of research instruments to explain the health-risk behaviors, added.
Also, why did you focus on university students? there should be some justification of methodological choices made.
Response: University students were the source population of this study
This paper would benefit greatly from some comparison between pre and post covid statistics.
Response: Local data pre-COVID is non-existence, although reference to other countries are included.

Reviewer 5 Report
This is an article that illustrates the consequences of Covid in areas that Western academic research does not deal with sufficiently. There is no theoretical innovation in the article, but rather an empirical one: illustrating the prices paid by a population that did not receive an appropriate health policy and that did not internalize among its population appropriate ways to deal with epidemics, including the Covid.
In order for the article to have a universal value and not just point to local empirical findings, I recommend that in the concluding chapter, the authors will point out that the article illustrates the connection between culture and (mental) health, and while doing so is another illustration that sometimes Culture may be a Recovery Barrier as illustrated in their article regarding dealing with the Covid epidemic. They should do so by referring to the article:
Lebel, U., "'Second Class Loss': Political Culture as a Recovery Barrier? – Israeli Families of Terrorist Casualties and their Struggle for National Honors, Recognition and Belonging", Death Studies, 38(1), 2014, 9-19 .
In addition - it is desirable that the authors make it clear that the quality of life of those dealing with epidemics will increase if public health professionals take on the role of internalizing what can be called "coping scripts" (nutritional, mental, social, communicative, etc.) More functional and healthier. This while referring to the pioneering article in the field of "Normative Scripts":
Lebel, U. and Masad, D. "Life Scripts, Counter Scripts and Online Videos: The Struggle of Religious-Nationalist Community Epistemic Authorities against Military Service for Women". Religions 12(9), 2021, 750.
Author Response
Thank you and we greatly appreciate your valuable time to review and comment on our humble manuscript. Relevant changes according to reviewers’ comments is highlighted in yellow in the revised manuscript, attached.
In order for the article to have a universal value and not just point to local empirical findings, I recommend that in the concluding chapter, the authors will point out that the article illustrates the connection between culture and (mental) health, and while doing so is another illustration that sometimes Culture may be a Recovery Barrier as illustrated in their article regarding dealing with the Covid epidemic. They should do so by referring to the article:
Lebel, U., "'Second Class Loss': Political Culture as a Recovery Barrier? – Israeli Families of Terrorist Casualties and their Struggle for National Honors, Recognition and Belonging", Death Studies, 38(1), 2014, 9-19 .
Response: Thank you for your valuable time to review and comment on our humble manuscript. All changes according to reviewers’ comments is highlighted in yellow in the revised manuscript. Indeed, cultural effects on recovery is important, discussion expanded on this point. Suggested article is also added.
In addition - it is desirable that the authors make it clear that the quality of life of those dealing with epidemics will increase if public health professionals take on the role of internalizing what can be called "coping scripts" (nutritional, mental, social, communicative, etc.) More functional and healthier. This while referring to the pioneering article in the field of "Normative Scripts":
Lebel, U. and Masad, D. "Life Scripts, Counter Scripts and Online Videos: The Struggle of Religious-Nationalist Community Epistemic Authorities against Military Service for Women". Religions 12(9), 2021, 750.
Response: Although we agree that internalizing mechanisms to deal with mental wellbeing is one effective way, it is not relevant and deviate from the main points the author would like to focus on. Thank you for the kind suggestions nonetheless.

Round 2
Reviewer 4 Report
The feedback has been addressed sufficiently.